# Evaluating Rhizobacterial Antagonists for Controlling *Cercospora beticola* and Promoting Growth in *Beta vulgaris*

**DOI:** 10.3390/microorganisms12040668

**Published:** 2024-03-27

**Authors:** Zakariae El Housni, Said Ezrari, Nabil Radouane, Abdessalem Tahiri, Abderrahman Ouijja, Khaoula Errafii, Mohamed Hijri

**Affiliations:** 1Laboratory of Biotechnology and Molecular Biology, Department of Biology, Faculty of Science, Moulay Ismail University, Zitoune, Meknès 50050, Morocco; z.elhousni@edu.umi.ac.ma (Z.E.H.); a.ouijja@edu.umi.ac.ma (A.O.); 2Phytopathology Unit, Department of Plant Protection, Ecole Nationale d’Agriculture de Meknès, BPS 40, Meknès 50001, Morocco; atahiri@enameknes.ac.ma; 3Microbiology Unit, Laboratory of Bioresources, Biotechnology, Ethnopharmacology and Health, Faculty of Medicine and Pharmacy Oujda, University Mohammed Premier, P.O. Box 724 Hay Al Quods, Oujda 60000, Morocco; said.ezrari@ump.ac.ma; 4African Genome Center, University Mohammed VI Polytechnic (UM6P), Lot 660, Hay Moulay Rachid, Ben Guerir 43150, Morocco; nabil.radouane@um6p.ma (N.R.); khaoula.errafii@um6p.ma (K.E.); 5Institut de Recherche en Biologie Végétale (IRBV), Département de Sciences Biologiques, Université de Montréal, Montréal, QC H1X 2B2, Canada

**Keywords:** biological control, *Cercospora beticola*, *Cercospora* leaf spot, plant growth-promoting rhizobacteria, sugar beet

## Abstract

*Cercospora beticola* Sacc. is an ascomycete pathogen that causes *Cercospora* leaf spot in sugar beets (*Beta vulgaris* L.) and other related crops. It can lead to significant yield losses if not effectively managed. This study aimed to assess rhizosphere bacteria from sugar beet soil as a biological control agent against *C. beticola* and evaluate their effect on *B. vulgaris*. Following a dual-culture screening, 18 bacteria exhibiting over 50% inhibition were selected, with 6 of them demonstrating more than 80% control. The bacteria were identified by sequencing the 16S rRNA gene, revealing 12 potential species belonging to 6 genera, including *Bacillus*, which was represented by 4 species. Additionally, the biochemical and molecular properties of the bacteria were characterized in depth, as well as plant growth promotion. PCR analysis of the genes responsible for producing antifungal metabolites revealed that 83%, 78%, 89%, and 56% of the selected bacteria possessed bacillomycin-, iturin-, fengycin-, and surfactin-encoding genes, respectively. Infrared spectroscopy analysis confirmed the presence of a lipopeptide structure in the bacterial supernatant filtrate. Subsequently, the bacteria were assessed for their effect on sugar beet plants in controlled conditions. The bacteria exhibited notable capabilities, promoting growth in both roots and shoots, resulting in significant increases in root length and weight and shoot length. A field experiment with four bacterial candidates demonstrated good performance against *C. beticola* compared to the difenoconazole fungicide. These bacteria played a significant role in disease control, achieving a maximum efficacy of 77.42%, slightly below the 88.51% efficacy attained with difenoconazole. Additional field trials are necessary to verify the protective and growth-promoting effects of these candidates, whether applied individually, combined in consortia, or integrated with chemical inputs in sugar beet crop production.

## 1. Introduction 

*Cercospora* leaf spot disease of sugar beet is one of the most significant foliar diseases in humid and temperate areas [1]. It is caused by the ascomycete fungus *Cercospora beticola* Sacc. (Family Mycosphaerellaceae) [2]. This pathogen plays a significant role in sugar beet production globally. It can decrease the tonnage yield by as much as 50% [3,4], concurrently affecting the amount of extractable sugar by increasing impurities [5,6].

The management of the disease involves several practices, including weed control, irrigation, and nitrogen fertilization. Breeding disease-resistant varieties is also a strategy employed to control disease severity; however, the resulting cultivars have been reported to be less productive than susceptible varieties [7]. Nonetheless, the latest varieties appear to perform better than the previous ones [8,9]. Despite the ongoing debate on the use of chemicals in pathogen control, this treatment remains the most effective way to manage the disease, especially for susceptible varieties [10]. 

The excessive use of fungicides against *C. beticola* has led to its development of acquired resistance to various chemical molecules, including benomyl [11], methyl thiophanate [12], tetraconazole, epoxyconazole, difenoconazole flutriafol [13,14], azoxystrobin, trifloxystrobin [15], and picoxystrobin [16]. Furthermore, several isolates are now capable of exhibiting multiple types of resistance [17]. To address this issue, biological control using different types of microorganisms (fungi, bacteria, and viruses) has emerged as a highly promising technology [18].

Biological control stands out as a promising approach to manage fungal disease resistance, gaining increasing attention as an environmentally friendly biotechnology. Bacteria are extensively utilized as biological control agents (BCAs) due to their notable effectiveness [19]. Their efficacy stems from their remarkable ability to produce a diverse range of antibiotic compounds. BCAs contribute to the reduction of fungal disease outbreaks through mechanisms such as competition for space and nutrients, the induction of plant systemic resistance, or the production of antifungal metabolites [19].

Lipopeptides, cyclic amphiphilic oligopeptides with a low molecular weight, are synthesized in BCAs by non-ribosomal peptide synthetases. They exhibit robust antimicrobial properties owing to their capacity to interact with cell membranes, leading to pore formation and membrane solubilization at higher concentrations. These lipopeptides are categorized into three primary families: Surfactins/lichenysins, iturins/bacillomycins/mycosubtilins, and fengycins/plipastatins. Bacteriocins, including subtilosin A, are ribosomally produced by various microorganisms, and they exhibit antimicrobial activity by forming membrane pores [20,21,22].

Both bacterial and fungal agents have demonstrated their potential as BCAs [23,24], addressing a wide spectrum of diseases, notably including the management of *C. beticola* [25,26]. Furthermore, rhizospheric bacteria are recognized for their abilities to promote plant growth, including traits such as alleviating soil nutrient deficiencies (e.g., nitrogen fixation, phosphorus solubilization, and improving iron uptake), synthesizing plant growth-promoting hormones, inhibiting ethylene production through 1-aminocyclopropane-1-carboxylate deaminase activity [27,28], and stimulating overall plant development [29,30], which extends to sugar beet cultivation [31,32].

In Morocco, multiple strains of *C. beticola* have been documented as resistant to multiple fungicide groups, as reported by El Housni et al. (2020) [14]. This study aimed to develop a biological alternative to manage this pathogen in sugar beet production. We formulated three hypotheses: (1) Bacteria obtained from the rhizosphere of healthy sugar beet plants will effectively control *Cercospora* leaf spot while promoting sugar beet growth; (2) bacteria carrying lipopeptide-encoding genes will demonstrate greater efficacy in inhibiting *C. beticola* growth; and (3) bacteria displaying a higher number of plant growth-promoting traits will enhance sugar beet growth. To test these hypotheses, we isolated, selected, and characterized several bacteria from sugar beet rhizospheric soil. These bacteria were assessed for their antagonistic potential against *C. beticola* both in vitro and in vivo. Furthermore, this study focused on characterizing the isolated bacteria, taking into account their biochemical and molecular attributes, along with evaluating their impacts as plant growth-promoting rhizobacteria.

## 2. Materials and Methods 

### 2.1. Fungal Material

The ascomycete fungus used in this present study was *C. beticola* isolate B9-1, which exhibited cross-resistance to the three main chemical families (methyl benzimidazole carbamate (DMI), demethylation inhibitors (DMI), and quinone outside inhibitors (QoIs)) [14]. This strain was originally isolated from a symptomatic sugar beet leaf during the 2017 growing season in Morocco. Before the experiments, the fungus isolate was subcultured on potato dextrose agar (PDA) and incubated for seven days in darkness at 25 °C.

### 2.2. Isolation of Bacteria from Sugar Beet Rhizosphere Soil

Sugar beet is planted as an autumn crop, sown in October and November, and harvested in June and July. Four months after sowing in the sugar beet regions of Gharb, Doukkala, and Tadla, plants that displayed no visible symptoms of *C. beticola* attacks and exhibited no foliar lesions from other pathogens (Appendix A) were selected. Bacteria were isolated from the rhizosphere soil of the healthy sugar beet plants, described in this context as the soil immediately surrounding the taproot of sugar beet. Eight samples were taken, with approximately 1 kg soil per sample collected from the vicinity of healthy sugar beet plants within a 15 cm diameter of the tap root and up to 50 cm deep. After collection, the soil from each plant was thoroughly mixed. The mixed soil was then placed in plastic bags and transported to the laboratory. 

One gram of each soil sample was mixed with 9 mL of sterile distilled water. After shaking, a series of dilutions from 10^−4^ to 10^−6^ were carried out [25]. A volume of 0.1 mL of each diluted suspension was incubated on potato dextrose agar (PDA) medium in Petri dishes at 28 °C for at least 48 h. The resulting bacterial colonies were subsequently transferred to new PDA Petri dishes until pure bacterial morphotype colonies were obtained (colonies with a homogeneous morphology). A single colony was chosen based on differences in color, shape, and size. A solid portion of pure culture of each bacterium was then placed in micro-tubes with sterile distilled water and stored at 4 °C [25]. BIOBAC^®^
*Bacillus subtilis* strain Y1336 (SAOAS, Casablanca, Morocco) was used as a reference. 

### 2.3. Screening of Antagonistic Activity among Rhizosphere Bacteria 

#### 2.3.1. Dual-Culture Setup for Confrontation Testing

The inhibitory potential of 50 bacterial isolates against *C. beticola* isolate B9-1 was assessed using the direct confrontation method on PDA medium, with four streaks made equidistant from the fungal colony [21]. The inhibition rate was calculated at day 7 according to the formula: *TI* = [(*Ct* − *Ci*)/*Ct*] × 100
where *TI* represents the growth inhibition rate of *C. beticola* (%), *Ct* is the diameter of *C. beticola* without rhizospheric bacteria (control), and *Ci* is the diameter of *C. beticola* in the presence of rhizospheric bacteria.

#### 2.3.2. Antibiotic Activity through Bacterial Supernatant Analysis

A 100 μL volume of each bacterial suspension (1 × 10^8^ CFU/mL) was inoculated into nutrient broth medium (NB) and then incubated in a rotary shaker at 25 °C for 6 days (130 rpm) in the dark. The mixture underwent centrifugation at 5500 rpm for 25 min, and the resulting supernatant was collected and filtered through a 0.22 µm pore size syringe filter. For the ‘Poison food technique’, the filtrate was added to PDA medium at 40 °C with a 10% filtrate concentration (50 mL filtrate/500 mL liquid PDA medium). Petri dishes containing the medium amended with the filtrate of rhizospheric bacteria were used to inoculate 7 mm diameter portions of *C. beticola* (obtained from a pure culture on PDA medium) taken with a crock borer. The incubation period lasted 7 days at 25 °C in complete darkness. Three replicates were carried out for each bacterial filtrate, and three replicates for the control (10% concentration of liquid NB medium added to the PDA instead of the bacterial supernatant). The Petri dishes were placed in the incubator at random locations.

The rate of indirect inhibition of *C. beticola* growth by the rhizosphere bacterial filtrates was calculated according to the following formula [33]: *GI* = [(*Cd* − *Ci*)/*Cd*] × 100
where:

*GI*: the growth inhibition rate of *C. beticola* (%);

*Cd*: the diameter of *C. beticola* without rhizosphere bacteria (control);

*Ci*: the diameter of *C. beticola* in the presence of rhizosphere bacteria.

*Bacillus subtilis* Y1336 was used as a reference strain (positive control).

### 2.4. Fourier Transform Infrared Spectroscopy (FTIR) Analysis

This analysis was conducted using an FTIR spectrophotometer (Shimadzu Model 8400S, Shimadzu, Noisiel, Marne-la-Vallée, France). A homogeneous mixture was prepared by dissolving one milliliter of the biosurfactant sample in potassium bromide pellets to create the samples (Merck, Rahway, NJ, USA). An integrated plotter was used to generate IR absorption spectra. The IR spectra were acquired with a resolution of 4 cm^−1^, covering a wavelength range of 450–4500 cm^−1^. The spectral measurements represented the average of 50 scans across the instrument’s complete operating range.

### 2.5. DNA Extraction and PCR Amplification of 16S rRNA Gene and Sanger Sequencing

DNA extraction from bacterial suspensions followed the protocol of Llop et al. (2000) [34]. Amplification of the 16S rRNA gene from the genomic DNA of bacterial isolates was carried out using the following pair of primers: 27F/1492R, which amplify a 1450 bp fragment [22]. The PCR reaction mixture was performed in a total volume of 25 μL containing PCR buffer, 2.5 μM of each primer, 1 U of *Taq* DNA polymerase (Bioline, London, UK), and 2.5 μL of DNA template. The following cycling conditions were used: initial denaturation at 96 °C for 4 min, followed by 35 cycles of denaturation at 96 °C for 10 s, annealing at 52 °C for 40 s, and elongation at 72 °C for 2 min; finally, an extension at 72 °C for 4 min. The PCR reactions were run in an Eppendorf Thermal Cycler (Eppendorf, Hambourg, Germany).

PCR reactions that showed amplification bands of the expected fragment of approximately 1.4 kb were sequenced using the Sanger dioxide method in the Stab Vida laboratory in Portugal. The obtained sequences were analyzed using DNAMAN software (version 6.0, Lynnon Biosoft, San Ramon, CA, USA), and then blasted at the NBCI website (http://blast.ncbi.nlm.nih.gov/Blast.cgi) accessed on 15 February 2024, using BLAST nucleotide search. The 16S rRNA gene sequences have been deposited in the GenBank database, along with their corresponding accession numbers.

### 2.6. PCR Amplifications of Lipopeptide-Encoding Genes

The genes encoding bacillomycin, fengycin, iturin, and surfactin were identified using the genomic DNA isolated from the 18 selected bacterial isolates. For each PCR amplification, a total volume of 25 µL of PCR mixture was used, which contained PCR buffer, 2.5 µM of each primer, 1U of *Taq* DNA polymerase (Thermo Fisher Scientific, Rabat, Morocco), and 2.5 µL of genomic DNA. Appendix A shows the specific primers designed to detect the target genes utilized for amplification. The PCR products were visualized on a 1.5% agarose gel stained with ethidium bromide (0.02 g/mL), observed under an ultraviolet light source, and digitally recorded.

### 2.7. Biochemical and Plant Growth Promotion Tests 

Different tests with three replicates were performed to detect various biochemical bacterial properties. Appendix A summarizes those tested with the adopted evaluation method. 

### 2.8. Effects of Bacterial Isolates on Sugar Beet Growth in Greenhouse Conditions 

The culture substrate used in this greenhouse trial consisted of 2/3 commercial peat and 1/3 sand. The substrate mixture was autoclaved twice at 121 °C for 30 min [35] in hermetically sealed plastic bags. A weight of 200 g of substrate per pot was used for testing the rhizospheric bacteria growth-promoting effect. Commercial sugar beet seeds of the Liberata cultivar (KWS, Casablanca, Morocco) were soaked in sterile distilled water for 3 h to remove the coating. Subsequently, the seeds were manually rubbed with a sterile glove and rinsed twice with sterile distilled water (Appendix A).

The bacteria isolated in this study were grown as pure cultures in yeast extract peptone (YEP) medium at 28 °C under agitation for 24 h. Subsequently, the bacterial cultures of each isolate were centrifuged at 5000 rpm for 20 min, and the supernatant was discarded. The resulting pellet was re-suspended in PBS at a concentration of 10^8^ CFU/mL, achieved by ensuring an OD 600 nm of 0.1.

Sugar beet seeds underwent disinfection with 70% ethanol for 1 min, followed by a single wash with sterile distilled water. Subsequently, they were soaked in a 5.25% sodium hypochlorite solution with a few drops of Tween-20 and then rinsed with sterile distilled water five times for 1 min [36]. The sterilized seeds were immersed in the bacterial solution for one hour at room temperature, while the control seeds were soaked in PBS alone.

Sugar beet seeds planted in pots containing sterilized substrate were placed in a controlled greenhouse with a temperature of 25 °C and relative humidity of 70%. The pots were placed in a completely randomized design with three replicates. After 30 days, the sugar beet plants were assessed for the development of both vegetative and root organs, considering the length and dry weight. The leaves and roots were measured with a ruler, while the dry weight was measured using a precision balance.

No chemical or organic fertilizers were added to the sugar beet plants, and irrigation was conducted once every three days with 60 mL of tap water per pot.

### 2.9. Effects of Four Bacterial Isolates on Sugar Beet Growth in the Field

Four bacterial isolates (BGH 1-6, BGH 2-2, BGH1-3, and BGH 2-7) displaying significant in vitro antagonism against *C. beticola* were chosen for a field trial. A suspension containing cells of these bacteria was utilized with a concentration of 1 × 10^8^ CFU/mL. The slurry application rate was 400 L/ha (Appendix A). 

The experiments were carried out in 2023 in the Gharb area of Province Sidi Slimane (34°22′12.9″ N 6°09′28.4″ W). The experimental design was a randomized complete block design with four replicates (Appendix A). Each field plot consisted of 8 m rows spaced 50 cm apart and planted with the sugar beet cultivar Liberta (KWS, Casablanca, Morocco), known for its medium tolerance to *Cercospora* leaf spot.

Treatment application occurred at the end of February, ensuring a uniform incidence of 64% through a backpack sprayer delivering a 400 L/ha spray. All test plots received the same recommended herbicide and insecticide treatments. Disease severity was monitored over four weeks, and the results were employed to calculate the area under the disease progression curve (AUDPC) [37]. Additionally, the leaf phytotoxicity of the bacterial treatment was evaluated.
$$AUDPC=\;\sum_{i=1}^{n}\left[\frac{Y_{i=1}+Y_{i}}{2}\right]\left[t_{i+1}-t_{i}\right],$$
where *Yi* represents the severity of the *i*th observation, *t_i_* denotes the time in days of the observation, and *n* is the total number of observations.

### 2.10. Statistical Analysis

The data were analyzed using ANOVA at a significance level of 5%, and in instances where the effect was deemed significant at *p* ≤ 0.001, a multiple comparison of the Duncan means was executed. Principal component analysis (PCA) was employed to explore the correlations among bacteria antagonism, lipopeptide-encoding genes, and hydraulic enzyme production. All analyses were conducted using XLSTAT Pro software, version 7.5.2.

## 3. Results

### 3.1. Isolating and Identifying Bacteria from the Rhizosphere Soil of Sugar Beet

A subset comprising 18 bacterial isolates underwent various screening tests. The identification of these isolates was carried out through 16S rRNA gene and Sanger sequencing, as detailed in Table 1. Sequencing enabled the identification of three isolates at the species level, with a sequence identity exceeding 99% (BGH1-6 was identified as *Pantoea agglomerans* and G1A as *Pseudomonas azotoformans*), and the isolate G4A matched *Pantoea* sp., even though the sequence length was approximately 550 bp. For the remaining 15 isolates, the percentage of sequence identity ranged between 91.12% and 97.84%, falling below the threshold for species identification [38]. 

The genus *Bacillus* dominated the isolated community with eight isolates, followed by *Pantoea* (four isolates), *Serratia* (three isolates), and *Enterobacter*, *Kosakonia*, and *Pseudomonas*, each represented by one isolate.

### 3.2. Antagonism against C. beticola in Confrontation Assay in PDA Medium and PDA Amended with Supernatant

We conducted two distinct yet complementary assays to evaluate the antagonistic impact of the 18 bacteria against *C. beticola*. In the confrontation assay on PDA medium, all of the isolates demonstrated significant (*F* = 4.340; *p* < 0.001) growth inhibition of more than 50%, compared to the control (Figure 1A). Notably, six isolates (BGH2-2, BGH2-5, BGH1-6, BGH 1-3, BGH 2-7, and G3d) exhibited a growth inhibitory effect exceeding 80%.

The second assay involved applying the supernatants of the 18 bacteria on PDA medium, which was subsequently used for the cultivation of *C. beticola*. The inhibitory effect of the selected bacteria was statistically significant (*F* = 2.112; *p* < 0.031) compared to the control, with four isolates (BGH1-3, G3c, BGH2-7, and BGH 4-1) showing more than 25% growth inhibition (Figure 1B).

### 3.3. PCR Detection of Lipopeptide Genes

PCR analysis revealed the existence of genes (*BamC, Itup, FenD*, and *Sfp*) encoding antifungal products, specifically leptopeptides (bacillomycin, iturin, fengycin, and surfactin). Table 2 summarizes the presence and absence of these genes in the bacterial isolates. The genes associated with lipopeptide production were found in varying proportions: 67%, 78%, 83%, and 89% for the *Sfp, Itup, BamC,* and *FenD* genes, respectively.

Four bacteria (G1d, G1b, BGH 1-3, and BGH 2-2) were identified to have all of the genes essential for lipopeptide production, each exhibiting different antagonistic capabilities in direct confrontation or through a filtrate against *C. beticola*. Among the six bacteria with an inhibition rate exceeding 80% in direct confrontation (Figure 1A), all possessed at least two lipopeptide production genes (*BamC* and *Itup*). Similarly, the four bacteria with an inhibition rate surpassing 25% by filtrate (Figure 1B) all harbored at least two genes responsible for lipopeptide production (*Itup* and *FenD*).

### 3.4. Biochemical and Plant Growth Promotion Tests 

The bacterial isolates underwent comprehensive characterization tests to unveil their functional traits. Table 3 shows the results of eight biochemical activities, including enzymatic assessments for cellulose, pectinase, protease, amylase, and chitinase, along with phosphate solubilization, indole acetic acid production, and hydrogen cyanide production. Among these findings, 56% of the bacteria exhibited proteolytic activity, 61% demonstrated phosphate solubilization, 33% displayed amylolytic activity, 28% showed cellulosic activity, and 28% had pectinase activity. Additionally, 17% demonstrated chitinase activity, while 78% displayed hydrogen cyanide production (Table 3).

The filtrate from the BGH 1-3 isolate’s supernatant, belonging to the *Serratia* genus, showed the best inhibition rate in both antagonistic assays. It underwent analysis using infrared (IR) spectroscopy. The observed peaks closely resembled the IR spectrum of bacterial lipopeptides. The presence of ester carbonyl groups (–C=O bond in –COOH) is indicated by the peak around 1049 cm^−1^ [39]. Additionally, the peak at approximately 1058 cm^−1^ suggests the existence of amide moieties in proteins [40], and the peaks between 1540 and 1645 cm^−1^ are characteristic of the –C=O amide I vibration [41].

Absorption in the 1500–1650 cm^−1^ range, typically absent in the FTIR spectra of glycolipid biosurfactants [42], was not observed. Furthermore, the peak between 1370 and 1470 (1463) indicates the presence of deformation and bending vibrations of the –C–CH2 and –C–CH3 groups in aliphatic chains. The two peaks in the 2850–2950 cm^−1^ region support the –CH stretching mode of the CH3 and CH groups in alkyl chains [41].

These findings suggest the presence of peptide groups in the bacterial supernatant filtrate, affirming the lipopeptide structure of the produced biosurfactant (Appendix A).

### 3.5. Effects of Bacterial Isolates on Sugar Beet Growth in Controlled Conditions 

The inoculation of the 18 bacterial isolates had a highly significant impact on sugar beet root length (*F* = 23.653; *p* < 0.0001) and root dry weight (*F* = 35.258; *p* < 0.0001). According to the Duncan test, three bacteria (BGH2-3, BGH 1-5, and BGH2-1) formed a distinct group with the best performance, while two others (G3c and G1a) showed a somewhat adverse effect on root growth (Figure 2A–C and Appendix A). Regarding the root dry weight, the Duncan test revealed various groups (Figure 2C). Seven bacterial isolates (G1b, BGH1-6, G2c, G3f, BGH2-3, BGH1-5, and BGH2-1) exhibited an enhanced effect exceeding 100%. Only G3d, BGH2-5, G3c, and G1a did not have a significant effect on the sugar beet root dry weight gain compared to the control (Figure 2 and Appendix A).

Bacterial inoculation also influenced root hair development (Appendix A and Figure 2D). Four bacterial isolates (BGH1-6, G1b, G2c, and G3f) had a highly positive effect on sugar beet root hair development, while the isolate G3c had a negative impact (Appendix A).

ANOVA indicated a highly significant effect (*F* = 23.65; *p* < 0.0001) of rhizosphere bacteria on increasing the length of the shoots of sugar beet. The Duncan test identified several groups (Figure 2E). Four bacterial isolates (BGH4-1, BGH1-5, BGH2-1, and BGH2-7) induced an increase in the length of the shoots by more than 50%, ranked from highest to lowest. Only G4a had a statistically insignificant effect compared to the untreated control (TNT).

Similarly, ANOVA showed a highly significant effect (*F* = 12.40; *p* < 0.0001) of rhizosphere bacteria on the weight of the sugar beet shoots. The Duncan test grouped the effect of the bacteria into several categories (Figure 2F). Six bacteria (BGH4-1, BGH1-5, BGH2-1, BGH2-7, G1B, and BGH1-6) induced an increase of more than 50% in the weight of the shoots, ranked from highest to lowest.

A comprehensive principal component analysis was conducted to examine the correlation between PGPR antagonism, including both direct confrontation assays and bacterial supernatant, and the bacterial ability to produce pectinase and cellulase. It is noteworthy that bacteria displaying higher rates of direct or indirect antagonism were more prominently linked to the production of hydrolytic enzymes rather than the presence of genes associated with antifungal metabolite production (Appendix A).

### 3.6. Effects of Four Bacterial Isolates on Sugar Beet Growth in the Field

No leaf phytotoxicity was found on sugar beet tested with the bacterial suspension. Four bacterial isolates were chosen for the field trial (BGH 2-7, BGH 1-3, BGH 2-2, and BGH 1-6), and the ANOVA revealed a highly significant impact of the treatment on the severity of *C. beticola* (*F* = 74.63; *p* < 0.0001). The Duncan test categorized the treatments into four distinct groups. The most effective treatment was difenoconazole, recording an efficacy of 88.5%, followed by three bacteria (BGH2-7, BGH 1-3, and BGH1-6) with respective efficacies of 77.4%, 73.8%, and 76.1%. BGH2-2 exhibited the lowest efficacy, with only 61.4% (Table 4).

## 4. Discussion

The aim of this study was to investigate the practical applications of bacterial bioinoculants with the ability to control *Cercospora* leaf spot disease in sugar beets while simultaneously enhancing plant growth and vigor. Bacteria were isolated from the rhizosphere of healthy sugar beets cultivated in different regions of Morocco. Among a subset of 18 bacterial strains subjected to various tests, including identification through 16S rDNA sequencing, the genus *Bacillus* was predominant, represented by eight isolates. This was followed by *Pantoea* (four isolates), *Serratia* (three isolates), and *Enterobacter, Kosakonia*, and *Pseudomonas*, each represented by one isolate. These findings are consistent with previous studies that have demonstrated the prevalence *Bacillus* species in conventional agricultural soils [43,44].

Sequencing of the 16S rRNA gene allowed for the identification of three isolates at the species level, with a sequence identity exceeding 99% (BGH1-6 identified as *Pantoea agglomerans* and G1A as *Pseudomonas azotoformans*), and the isolate G4A matched *Pantoea* sp., although the sequence length was approximately 550 bp for these isolates. The expected fragment size of the primers utilized for amplifying the 16S rRNA gene is approximately 1400 bp. However, for shorter fragments, the sequencing quality was compromised, with either poor sequencing results or successful sequencing with only one primer instead of both forward and reverse primers. For the remaining 15 isolates, the percentage of sequence identity ranged between 91.12% and 97.84%, falling below the threshold for species identification, proposed as 98.7% [45]. This percentage falls into the range of genus identification, which is 94.9% [45]. Out of the 18 bacterial isolates, 16 could be potential new species, including G4A (*Pantoea* sp.). Further investigations, such as whole-genome sequencing, phylogenetic analysis, and DNA–DNA hybridization, are required to determine the taxonomic position of these isolates or to describe new species.

Moreover, we conducted two antagonist tests, including a direct confrontation test on PDA medium and an indirect test involving the supplementation of the bacterial culture supernatant on PDA medium where *C. beticola* was cultivated, thereby revealing the potential of the isolated bacteria to inhibit the growth of *C. beticola*. A majority of the bacterial isolates exhibited a significant antagonistic effect, particularly in dual confrontation cultures against *C. beticola*. Isolates with an antagonism rate exceeding 80% included BGH2-2 (*Serratia* sp.), BGH 1-3 (*Serratia* sp.), BGH2-5 (*Pantoea* sp.), BGH1-6 (*Pantoea* sp.), BGH 2-7 (*Bacillus* sp.), and G3d (*Bacillus* sp.). These bacteria are recognized for their ability to inhibit various pathogens [46,47,48,49,50]. Notably, while only *Bacillua* spp. [26,38,50,51] have been previously tested against *C. beticola*, our study is the first to investigate the efficacy of bacterial taxa, such as *Pantoea, Kosakonia, Bacillus*, *Serratia*, and *Pseudomonas azotoformans,* as biological control agents against *C. beticola*, demonstrating in vitro radial inhibition exceeding 50%.

Unlike other studies, our in vitro antagonism tests, utilizing the addition of bacterial culture supernatant, yielded relatively lower inhibition rates, not surpassing 29%. Interestingly, species affiliated with the *Bacillus* genus demonstrated inhibition rates exceeding 50% [25], despite PCR analysis confirming the presence of several genes associated with lipopeptide production in the other tested bacteria.

Indeed, upon comparing the growth inhibition results from both direct and indirect confrontations with the presence of genes encoding lipopeptides (bacillomycin, fengycin, iturin, and surfactin) (Appendix A), we found that the presence of these genes did not necessarily correlate with optimal performance in inhibiting the growth of *C. beticola*. This was exemplified by the bacterial isolate G1d (*Bacillus* sp.), which possessed all four lipopeptide-encoding genes yet demonstrated a direct antagonism of 51% and 0% in the indirect antagonism test. This finding does not support our second hypothesis, which suggests that bacteria carrying lipopeptide-encoding genes would exhibit greater effectiveness in inhibiting *C. beticola* growth. Conversely, the top six bacterial isolates in direct confrontation all had at least the bacillomycin-encoding gene, while the four top-performing bacteria in the indirect confrontation possessed at least two genes encoding for fengycin and iturin. Specific conditions may be necessary for each bacterial taxon to express these genes. The biosynthesis of surfactin and fengycin is influenced by the culture medium, with factors such as nutrient availability, sugar type [52], and the presence of oxidative [53] or diamide [54] agents playing a role in modulating surfactin production. This implies that testing for antagonism through lipopeptide extraction via supernatant harvesting based on only one combination of culture media and temperature in in vitro tests might be underestimating the potential antagonism of these bacteria.

Some bacteria possess the ability to produce lipopeptides, recognized for their significant impact on inhibiting fungi, and chitinase, which enables antagonistic bacteria to degrade the cell walls of fungi [21]. Eleven of our bacterial isolates, displaying diverse levels of antagonism (both direct and indirect), exhibited a positive association with three lipopeptides: bacillomycin, fengycin, and iturin encoding genes, along with the capacity to produce chitinase.

The field experiment demonstrated that the tested isolates delivered effective fungal growth inhibition control, with efficacy ranging from 61% to 77%. These findings align with those reported by Arzanlou et al. [25] in the context of post-infectious inoculation. Nevertheless, the chemical treatment using difenoconazole proved to be more effective, achieving an efficacy of 88.5%.

The in vivo experiment involving the four bacterial isolates (BGH 2-7, BGH 1-3, BGH 2-2, and BGH 1-6) validated the efficacy of these candidates in reducing *C. beticola* damage on sugar beet plants. They are potential candidates for biological control agents.

A sustainable approach involving biological solutions appears to be effective against *C. beticola*, particularly in preventive measures. This has been demonstrated with *Pythium oligandrum*, a mycoparasite used to control fungal diseases, showing a remarkable preventive efficacy of 87% [55]. Examining the possible correlations among various bacterial species [56] or fungal-bacterial antagonists [57,58] holds promise, particularly when these combinations demonstrate additive or synergistic effects. Another area of research involves delving into the interaction between antagonistic bacteria and fungicides [59]. Moreover, the identification of factors that induce antagonist-induced systemic resistance provides opportunities to strengthen plant resistance or integrate it with other methods to control specific pathogens [60].

In addition to their role as microbial control agents for managing plant diseases, stimulating plant growth, and enhancing overall performance and yield, plant growth-promoting microbes play an important role in suppressing plant diseases [56]. They achieve this by producing inhibitory chemicals and inducing immune responses in plants against phytopathogens [56]. As biofertilizers and biopesticides, plant growth-promoting microbes are regarded as a viable and economically attractive approach for sustainable agriculture.

The bacterial isolates studied had several biochemical traits that can have a promotional effect on sugar beet cultivation. Bacteria can improve plant development by making nutrients, such as phosphate, available in the soil (61% of the bacteria tested), producing plant hormones (67% of the bacteria tested produced IAA), or degrading the walls of plant pathogens (28% each produced cellulase or pectinase).

Bacteria belonging to *Bacillus* sp., *Pantoea* sp., *Serratia* sp., and *Kosakonia* sp. have been successfully demonstrated to have plant growth promotion effects on sugar beet crops [32,57]. Meanwhile, others, such as *Bacillus subtilis*, *Bacillus amyloliquefaciens*, *Bacillus siamensis*, *Kosakonia cowanii*, and *Serratia nematodiphila*, have shown positive effects on the growth of various plants [58,59,60,61,62,63].

In this study, six bacteria (*Serratia nematodiphila* BGH 2-3, *Pantoea agglomerans* BGH 1-5, *Pantoea conspicua* BGH 2-1, *Bacillus subtilis* G3f, *Bacillus halotolerans*, and *Bacillus amyloliquefaciens* G2c) demonstrated the ability to increase root length and weight by over 100%. This increase could be attributed to the inhibition of root ethylene through an ACC deaminase effect [64]. These results are consistent with our third hypothesis, suggesting that bacteria exhibiting a greater abundance of certain traits will promote sugar beets.

Only four bacterial isolates (BGH 4-1, BGH 1-5, BGH 2-1, and BGH 2-7) were able to increase the length of the plant shoots by more than 50%. In terms of their total weight, 17 out of 18 bacteria had a positive effect on increasing the total plant weight, with only nine bacteria showing the ability to increase the weight by more than 50%.

When comparing the impact of bacterial inoculation on the development of plant roots and shoots, we observed that the greatest increase in root length did not necessarily correlate with the increase in root dry weight (e.g., BGH 2-3 versus BGH 4-1) or an increase in root hairs (G1b versus G3c). However, for the shoots, a positive proportional relationship existed between the increase in length and the increase in dry weight. One possibility is that the significance of root hair development may explain the variation in dry weight. Surprisingly, beetroot plants with a root dry weight increase of over 50% had less developed root hairs than the control (e.g., BGH 4-1 and BGH 1-5).

Several rhizobial bacteria were found to exhibit an inhibitory effect on the primary root in favor of lateral roots and root hairs [65]. Manipulation of these bacterial-mediated alterations in root traits paves the way toward sustainable crop production, with some bacteria positively influencing root hair development, possibly through auxin-independent mechanisms [66]. Another avenue where rhizobial bacteria can contribute to improved agriculture is in the induction of systemic resistance, aiding in the control of plant pests in general [67]. 

## 5. Conclusions

The bacterial isolates examined and evaluated in this study exhibited multiple biochemical characteristics that led to a plant growth-promoting effect on *B. vulgaris* plants. There is a high likelihood of their utility as a biological control agent against *C. beticola*, as confirmed by both in vitro and in vivo trials. These antagonist bacteria, derived from the soil of healthy sugar beet plants, support our hypothesis and align with the view that the selection of microorganisms as biological control agents should involve screening from local niches and application in the same environment to achieve the desired benefits.

The positive outcomes of this study mark a crucial step, offering new possibilities for the development of biocontrol strategies to manage *Cercospora* leaf spot resistance against various fungicide groups. Consequently, four distinct antagonist bacterial isolates (*Serratia* sp. (BGH2-2), *Pantoea* sp. (BGH1-6), *Serratia* sp. (BGH 1-3), and *Serratia* sp. (BGH 2-7)), either individually or in consortia, are proposed for controlling and preventing damage caused by *C. beticola*. However, there is still a need to identify synergistic combinations among different bacterial isolates and determine the best compatibilities between bacterial isolates and sugar beet varieties to optimize the results in promoting cultivation and resistance against diverse adversaries. Additionally, exploring the combined use of fungicides and antagonistic bacteria in the field, particularly in addressing resistance issues, presents another avenue for future investigation.

The demonstrated effects of these rhizobacteria are highly significant and offer the serious possibility of enhancing the growth of *B. vulgaris*. Further tests must be conducted to determine the ideal combinations of different bacterial isolates, soils, and sugar beet varieties.

## Figures and Tables

**Figure 1 microorganisms-12-00668-f001:**
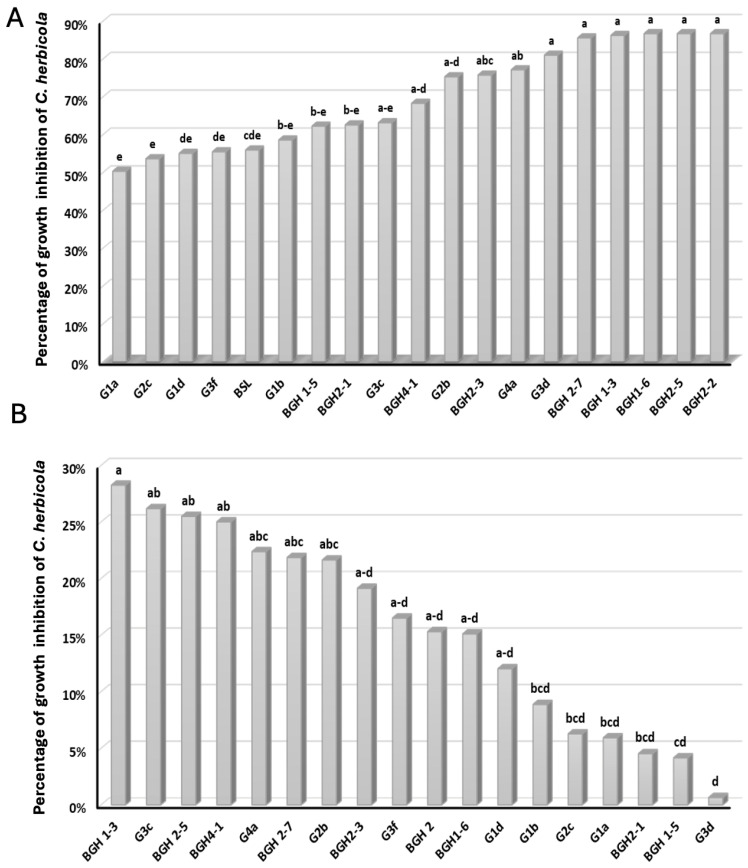
(**A**) Percentage of growth inhibition of *C. beticola* on PDA medium in dual culture, involving 18 bacterial isolates with *B. subtilis* Y1336 as a reference. (**B**) Percentage of growth inhibition of *C. beticola* on PDA medium implemented by the supernatants of the 18 bacterial isolates. BSL denotes *B. subtilis* Y1336. Different letters in bars (a, b, bc, etc.) denote significant differences between the percentages of inhibition of the tested isolates according to the Duncan test (*p* < 0.05).

**Figure 2 microorganisms-12-00668-f002:**
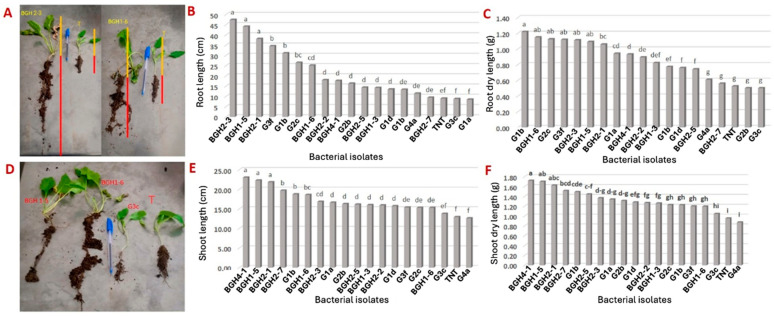
(**A**) Impact of bacterial inoculation on the growth of sugar beet, illustrating the roots (depicted in red) and shoots (depicted in yellow) of sugar beet plants (T refers to the untreated control). (**B**) Influence of bacterial inoculation on the growth of sugar beet root length (TNT refers to the control). (**C**) Influence of bacterial inoculation on sugar beet root dry weight (TNT refers to the control). (**D**) Examples showcasing the impact of bacterial inoculation on root hair development (T refers to the control). (**E**) Impact of bacterial inoculation on sugar beet shoot length growth (TNT refers to the control). (**F**) Impact of bacterial inoculation on sugar beet shoot dry weight (TNT refers to the control). Different letters in bars (a, b, bc, etc.) denote a significant difference between the percentages of inhibition of the tested isolates according to the Duncan test (*p* < 0.05).

**Table 1 microorganisms-12-00668-t001:** Sequencing of bacteria based on 16S rRNA gene with accession number.

Strain ID	Accession Number	Length (bp)	Coverage (%)	Identity (%)	Closest Taxon (Accession Number)
BGH2-7	MT256074	1501	99	95.56	*Bacillus vallismortis* (FJ386541)
BGH4-1	MW002558	559	100	96.08	*B. halotolerans* (MT271912)
G1B	MT256075	1477	99	97.35	*B. subtilis* (KP876486)
G2B	MT256077	1495	99	96.45	*B. halotolerans* (MF417800)
G3C	MT256076	1483	99	96.46	*B. amyloliquefaciens* (PP125657)
G1D	MW002221	1019	99	97.84	*B. subtilis* (KY818937)
G3F	MT254817	1524	99	94.72	*B. subtilis* (OM978656)
G3D	MT256072	1500	98	95.29	*B. subtilis* (KY652939)
BGH2-3	MW086541	536	99	95.86	*Enterobacter* sp. (JX103562)
BGH1-5	MW092092	1000	100	93.47	*Kosakonia cowanii MG871199*
BGH2-5 *	MT254758	1513	98	94.16	*Pantoea agglomerans* MZ647535
BGH1-6 *	MT254751	576	100	100	*P. agglomerans* (OQ202156)
BGH2-1	MT254818	903	98	91.12	*P. conspicua* (MW568057)
G4A	MW092005	555	99	99.29	*Pantoea* sp. (JN853255)
G1A	MW079530	539	100	99.63	*Pseudomonas azotoformans* (MK883209)
BGH1-3	MW079843	559	100	97.50	*Serratia liquefaciens* (MN326772)
BGH2-2	MW008870	301	99	97.00	*S. nematodiphila* (MH669373)
G2C	MW008604	645	99	96.57	*S. nematodiphila* (MN691578)

* These isolates are closely related, showing more than 99% sequence similarity.

**Table 2 microorganisms-12-00668-t002:** PCR amplification of antimicrobial lipopeptide-encoding genes was performed on the DNA extracted from the 18 bacterial strains. The symbol (+) indicates a positive amplification band of the expected size, whereas the symbol (-) denotes the absence of amplification.

Bacterial Isolates	Leptopeptide-Encoding Genes			
	*BamC*	*Itup*	*FenD*	*Sfp*
G1A	+	+	+	-
G2C	+	-	+	+
G1D *	+	+	+	+
G3F	+	+	-	+
G1B *	+	+	+	+
BGH 1-5	+	-	+	-
BGH2-1	+	+	+	+
G3C	-	+	+	-
BGH4-1	-	+	+	+
G2B	-	+	+	+
BGH2-3	+	-	+	-
G4A	+	+	+	-
G3D	+	-	+	+
BGH 2-7	+	+	+	-
BGH 1-3 *	+	+	+	+
BGH1-6	+	+	-	-
BGH2-2 *	+	+	+	+
BGH2-5	+	+	+	-

* Indicates the bacterial strains that demonstrated positive PCR amplification for all examined lipopeptide-encoding genes.

**Table 3 microorganisms-12-00668-t003:** Biochemical activity assessments of the bacteria. ICL: cellulose index; IPC: pectinase index; IPR: protease index; ISP: phosphate solubilization; IAM: amylase index; ICH: chitinase index; AIA: indole acetic acid production; HCN: hydrogen cyanide production.

Bacteria Isolates	ICL^a^	IPC ^a^	IPR ^a^	ISP ^a^	IAM ^a^	ICH ^a^	AIA	HCN
G2c	0	1.37 ± 0.08	4.12 ± 0.28	1.02 ± 0.16	1.26 ± 0.06	0	+++	++
BGH 1-5	0	0	1.17 ± 0.41	1.49 ± 0.16	0	5.21 ± 0.71	+++	−
BGH 2-1	0	0	0	0	0	1.94 ± 0.47	+++	+++
BGH 2-5	0	0	0	1.01 ± 0.43	0	0	++	+
BGH 1-6	1.5 ± 0.04	1.98 ± 0.79	0	1.16 ± 0.62	0	0	+	+
G3f	0	0	3.76 ± 0.22	1.37 ± 0.08	0	0	+	−
G2b	0	0	0	1.40 ± 0.12	0	0	+	++
BGH 2-2	0	0	1.21 ± 0.46	1.25 ± 0.19	0	0	+	++
BGH 1-3	1.61 ± 0.09	1.36 ± 0.34	4.12 ± 0.30	0	1.28 ± 0.04	1.53 ± 0.01	+	+++
G3d	1.57 ± 0.06	0	3.09 ± 1.45	0	0	0	+	+
G4a	1.54 ± 0.01	1.58 ± 0.11	1.82 ± 0.10	0	1.14 ± 0.00	0	+	+
BGH 2-7	0	2.03 ± 0.24	5.0 ± 0.21	1.41 ± 0.26	1.06 ± 0.00	0	+	++
G1b	0	0	0	0	1.19 ± 0.01	0	−	−
BGH 2-3	0	0	0	1.55 ± 0.01	0	0	−	+
BGH 4-1	0	0	4.85 ± 0.10	1.14 ± 0.20	1.24 ± 0.05	0	−	+++
G1d	0	0	0	1.19 ± 0.47	0	0	−	+
G1a	0	0	0	0	0	0	−	+
G3c	1.62 ± 0.05	0	1.42 ± 0.11	0	0	0	−	−

(0) and (−): negatives for the trait; (+++), (++), and (+): from strong to weak response for the trait, respectively. ^a^ Values are the mean of three replicates of independent assays ± standard error. Units are in mm.

**Table 4 microorganisms-12-00668-t004:** The impact of four bacterial isolates and difenoconazole on *Cercospora* leaf spot in sugar beets was assessed by measuring the area under the disease progression curve (AUDPC) in both the treated and control plots, along with determining the percentage of efficacy.

Treatments	Identity (Closest BLAST Match) ^a^	AUDPC ^b^	Efficiency (%)	Significance
BGH 2-7	*Bacillus vallismortis*	10.31 ± 0.12	77.42%	b
BGH 1-3	*Serratia liquefaciens*	11.94 ± 0.14	73.86%	b
BGH 2-2	*Serratia nematodiphila*	17.61 ± 0.24	61.45%	c
BGH 1-6	*Pantoea agglomerans*	10.91 ± 0.33	76.10%	b
Difenoconazole		5.25 ± 0.43	88.51%	a
Control		45.68 ± 0.96		d

^a^ Based on the percentage of sequence identity obtained by BLAST search (Table 1). ^b^ AUDPC means the area under the disease progression curve. Different letters (a, b, c, and d) denote a significant difference between AUDPC of the evaluated treatments according to the Duncan test (*p* < 0.05).

## Data Availability

The 16S rRNA gene sequences have been deposited at GenBank under the following accessions: MT256072, MT256074 to MT256077, MT254751, MT254758, MT254817, MT254818, MW002221, MW002558, MW008604, MW008870, MW079843, MW079530, MW092005, MW092092, and MW086541.

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
