# Peer review of "Evaluating Rhizobacterial Antagonists for Controlling Cercospora beticola and Promoting Growth in Beta vulgaris"

_microorganisms, 2024, doi:10.3390/microorganisms12040668_

Round 1

Reviewer 1 Report

Comments and Suggestions for Authors

An article submitted to Microorganisms by Zakariae El Housni et al. (the corresponding author is not specified) is devoted to searching for soil bacteria that benefit the growth and development of Beta vulgaris and possess antagonistic activity against Cercospora beticola. 

It is unclear from the text of the work what hypothesis the authors of the manuscript tested - what was primary, the antagonistic effect against Cercospora beticola or plant growth promotion. At the beginning of the work, it is necessary to decide which strain is considered promising and will be studied in the future and which is not - one that has a beneficial effect on the crop or which fights the pathogen. All experimental design (and associated statistical analysis) must proceed from the point of view of this assumption (hypothesis).

In its present form, the material in the article is presented in an extremely chaotic manner.

In addition to this significant comment, the reviewer asks the authors to answer several questions and correct several issues.

1) What do the letters a, ab, de and others mean on the graphs Fig. 1A, Fig. 1B? Why are the results of statistical analysis not provided for these data? Likewise, the statistical analysis results are not presented in Fig. 2B, C, E, F, and the symbols a, b, bc... are not explained on the graphs.

2) In Table 2, the authors indicate the presence of genes in the samples. PCR is used. But how do the authors know that these genes are expressed? If the RT-qPCR method is not used, discussing any biochemical or physiological effect of these genes is premature.

The reviewer found several typos and inaccuracies in the work; there are probably many more; the authors should pay attention to this and carefully proofread the manuscript again.

Line 23 - Should 16S rDNA be used?

line 178 - "2.5 L of genomic DNA" - whether mkl should be used?

Line 365 - Should "pots" be used instead of "plots"?

Line 367 - Should "by" be used instead of "buy"?

Comments on the Quality of English Language

The article contains multiple typos; editing of the manuscript by a professional English speaker is recommended.

Author Response

Dear Reviewer#1

We found your comments very helpful to make this major revision of the manuscript that we feel answered all the criticisms and makes it a much more solid and rigorous manuscript. We are very grateful to you for taking the time to think about this work seriously and provide us with the opportunity to present a more concise paper. In view of the comments, we have modified the text and added new references to address your comments. All changes are shown in red.

Answers point by point:

An article submitted to Microorganisms by Zakariae El Housni et al. (the corresponding author is not specified) is devoted to searching for soil bacteria that benefit the growth and development of Beta vulgaris and possess antagonistic activity against Cercospora beticola. 

Answer: We added the the corresponding author.

It is unclear from the text of the work what hypothesis the authors of the manuscript tested - what was primary, the antagonistic effect against Cercospora beticola or plant growth promotion. At the beginning of the work, it is necessary to decide which strain is considered promising and will be studied in the future and which is not - one that has a beneficial effect on the crop or which fights the pathogen. All experimental design (and associated statistical analysis) must proceed from the point of view of this assumption (hypothesis).

Answer: We appreciate the reviewer's feedback. As you will see in this revision, we outlined three hypotheses in the final paragraph of the Introduction. Additionally, we included text in the Discussion section where we addressed whether these hypotheses were confirmed or rejected.

In its present form, the material in the article is presented in an extremely chaotic manner.

Answer: We apologize for any inconvenience caused. Given the comprehensive nature of this study, various techniques were employed, making it challenging to integrate all methodologies seamlessly. In this revision, we made efforts to enhance both the structure and language of the manuscript, particularly Mat&Meth section.

In addition to this significant comment, the reviewer asks the authors to answer several questions and correct several issues.

1)What do the letters a, ab, de and others mean on the graphs Fig. 1A, Fig. 1B? Why are the results of statistical analysis not provided for these data? Likewise, the statistical analysis results are not presented in Fig. 2B, C, E, F, and the symbols a, b, bc... are not explained on the graphs.

Answer: We express our gratitude to the reviewer for providing valuable feedback. Significant differences among groups are denoted by different letters in the bars (a, b, bc, etc.), as determined by the Duncan test (p < 0.05). Please refer to the revised version of the manuscript for further details.

2) In Table 2, the authors indicate the presence of genes in the samples. PCR is used. But how do the authors know that these genes are expressed? If the RT-qPCR method is not used, discussing any biochemical or physiological effect of these genes is premature.

Answer: We appreciate the reviewer's suggestions. RT-PCR could have been ideal for monitoring the gene expression of these candidate genes. However, at the time of the experiment, we lacked both the necessary reagents and the instrument, which is often the case in developing countries. Therefore, we opted for PCR amplification to gain insights into the presence or absence of the genes.

Line 23 - Should 16S rDNA be used?

Answer: It is important to specify the region that underwent sequencing for the identification of the various bacterial species for non-microbiologist readers.

line 178 - "2.5 L of genomic DNA" - whether mkl should be used?

Answer: We appreciate the reviewer's comments. The correction has been made accordingly in the manuscript, where 2.5 µl of genomic DNA was specified.

Line 365 - Should "pots" be used instead of "plots"?

Answer: This was corrected.

Line 367 - Should "by" be used instead of "buy"?

Answer: Corrected.

The article contains multiple typos; editing of the manuscript by a professional English speaker is recommended.

Answer: We appreciate the reviewer's comment and recognize the significance of maintaining high standards in scientific writing. Your feedback has been taken seriously, and we have made efforts to improve the overall quality of the manuscript accordingly.

Reviewer 2 Report

Comments and Suggestions for Authors

The manuscript addressed selection of rhizobacterial biocontrol agents for Cercospora beticola and growth promotion in Sugar Beets. The manuscript is generally good However, the following points should be consider: 

1- The title should be revised as follows "Screening Rhizobacterial Antagonists for Biocontrol of Cercospora beticola and Growth Promotion in Beta vulgaris"

2-The novelty and motivation of the work should be expanded in the introduction

3- L171-173 mentioned the phylogenetic analysis but it seems not presented in results and discussion sections. 

4- In Table (1) use abbreviation of scientific names (e.g., Bacillus ).

5- The Entire manuscript should be revised making sure all scientific names should be italic (e.g.., L371, 387,406, 680)

6-L388,  How can phylogenetic analysis be involved in describing a novel species?

7- In Figure 1 and 2, statistical significance should be mentioned in the legends.

8- Another important point which is the short sequence length (500-600 bp) for some isolates may have impacted its accurate identification. Please elaborate more on this point taking into account the primers used in amplification of the 16S rRNA gene is 27F/1492R.

9- L461-465, Did the authors test the seven bacterial isolates for in vitro production of ACC deaminase? I suggest citing relevant references for the seven named bacterial strains for the plant growth promoting traits. 

10-I suggest explaining the significant plant growth stimulation of the bacterial strains considering the biochemical characteristics mentioned in Table S2 e.g phosphate solubilization 

Comments on the Quality of English Language

Minor editing of English language required

Author Response

Dear Reviewer#2,

We found your comments very helpful to make this major revision of the manuscript that we feel answered all the criticisms and makes it a much more solid and rigorous manuscript. We are very grateful to you for taking the time to think about this work seriously and provide us with the opportunity to present a more concise paper. In view of the comments, we have modified the text and added new references to address your comments. All changes are shown in red.

Answers point by point:

The manuscript addressed selection of rhizobacterial biocontrol agents for Cercospora beticola and growth promotion in Sugar Beets. The manuscript is generally good However, the following points should be consider: 

  • The title should be revised as follows "Screening Rhizobacterial Antagonists for Biocontrol of Cercospora beticola and Growth Promotion in Beta vulgaris"

Answer: we agree with the reviewer’s comment, we chawed the title accordingly.

2-The novelty and motivation of the work should be expanded in the introduction

Answer: We value the reviewer's input; we have outlined hypotheses in the concluding section of the introduction, highlighting the study's originality.

3-L171-173 mentioned the phylogenetic analysis but it seems not presented in results and discussion sections. 

Answer: We thank the reviewer for his comment, we have Deleted the part of phylogenetic analysis as we didn’t represent it in the result section.

4- In Table (1) use abbreviation of scientific names (e.g., Bacillus ).

Answer: We agree, modifications were made to the text.

5- The Entire manuscript should be revised making sure all scientific names should be italic (e.g.., L371, 387,406, 680)

Answer: We agree, modifications were made to the text.

6-L388, How can phylogenetic analysis be involved in describing a novel species?

Answer: Phylogenetic analysis aids in establishing the taxonomic status of the investigated isolates and facilitates the identification of potential new species. However, further analyses are essential for the formal description of new species, as discussed.

7- In Figure 1 and 2, statistical significance should be mentioned in the legends.

Answer: We agreed with the reviewer comment, Different letters in bars (a, b, bc, etc.) denote significant difference according to Duncan test (p < 0.05). Please see the modified version of the manuscript  

8- Another important point which is the short sequence length (500-600 bp) for some isolates may have impacted its accurate identification. Please elaborate more on this point taking into account the primers used in amplification of the 16S rRNA gene is 27F/1492R.

Answer: We agree with the reviewer’s comment, We added the following sentence ‘The expected fragment size of the primers utilized for amplifying the 16S rRNA gene is approximately 1400 bp. However, for shorter fragments, sequencing quality was compromised, with either poor sequencing results or successful sequencing of only one primer instead of both forward and reverse primers’.

9- L461-465, Did the authors test the seven bacterial isolates for in vitro production of ACC deaminase? I suggest citing relevant references for the seven named bacterial strains for the plant growth promoting traits. 

Answer: We did not employ the ACC deaminase test in the study; rather, we speculated that it could potentially serve as one of the explanations for the observed root inhibition effect. Additional references supporting this assumption have been included in the manuscript.

10-I suggest explaining the significant plant growth stimulation of the bacterial strains considering the biochemical characteristics mentioned in Table S2 e.g phosphate solubilization 

Answer: A paragraph was added discussing the suggestion made by the reviewer, please to see the revised version.

Comments on the Quality of English Language

Minor editing of English language required.

 Answer: we thank the reviewer for his comment, we understand the importance of adhering to high standards of scientific writing, and we have taken your feedback seriously to enhance the overall quality of the manuscript.

Reviewer 3 Report

Comments and Suggestions for Authors Dear authors,   The manuscript "Identifying Antagonistic Rhizobacterial strains with Biocontrol Potential Against Cercospora beticola and Assessing Their Growth-Promoting Effects on Beta vulgaris", in my opinion, has condition of being published in the journal Microorganisms, as it makes a significant contribution to knowledge in its area of science, and only a few corrections should be made: - The Abstract is too long and contains too much information that is unnecessary in this section. Please delete "The disease is characterized by circular lesions with an ash-gray center and dark brown to purple margins symptoms on the leaves." (L.18-19); "Subsequently, the bacteria were assessed for their effect on sugar beet plants in controlled conditions." (L.30-31); "Additional field trials are necessary to verify the protective and growth-promoting effects of these candidates, whether applied individually, combined in consortia, or integrated with chemical inputs in sugar beet crop production." (L.35-38). In fact, this last sentence (L.35-38) should be in the Discussion. - Finish the Abstract conclusively. - L.233: Why is the acronym for Principal Component Analysis not PCA? - L.330 and 336: standardize Duncan test. Why did you sometimes write DUNCAN? - L.399: correct for Bacillus - General: standardize throughout the manuscript, including supplementary material, Bacillus sp. or Bacillus spp.? _ L.493 (Conclusions): This entire section should be improved and focus on the results obtained. Nothing was mentioned about the promotion of growth in sugar beet plants by the bacterial isolates. Please improve the Conclusions accordingly. -L.504: sp. not italicized. Also review the entire text.                    

Author Response

Dear Reviewer#3,

We found your comments very helpful to make this major revision of the manuscript that we feel answered all the criticisms and makes it a much more solid and rigorous manuscript. We are very grateful to you for taking the time to think about this work seriously and provide us with the opportunity to present a more concise paper. In view of the comments, we have modified the text and added new references to address your comments. All changes are shown in red.

Answers point by point:

Dear authors, the manuscript "Identifying Antagonistic Rhizobacterial strains with Biocontrol Potential Against Cercospora beticola and Assessing Their Growth-Promoting Effects on Beta vulgaris", in my opinion, has condition of being published in the journal Microorganisms, as it makes a significant contribution to knowledge in its area of science, and only a few corrections should be made:

- The Abstract is too long and contains too much information that is unnecessary in this section. Please delete "The disease is characterized by circular lesions with an ash-gray center and dark brown to purple margins symptoms on the leaves.

Answer:  We agree, see the modified version of the manuscript  

" (L.18-19); "Subsequently, the bacteria were assessed for their effect on sugar beet plants in controlled conditions.

" (L.30-31); "Additional field trials are necessary to verify the protective and growth-promoting effects of these candidates, whether applied individually, combined in consortia, or integrated with chemical inputs in sugar beet crop production."  (L.35-38). In fact, this last sentence (L.35-38) should be in the Discussion.  - Finish the Abstract conclusively.

Answer: We appreciate the reviewer's comment, and several modifications have been made accordingly. The abstract has been restructured for clarity.

- L.233: Why is the acronym for Principal Component Analysis not PCA?

Answer: we agree, see the modified version of the manuscript. 

- L.330 and 336: standardize Duncan test. Why did you sometimes write DUNCAN?

Answer: We agree, see the modified version of the manuscript.

- L.399: correct for Bacillus - General: standardize throughout the manuscript, including supplementary material, Bacillus sp. or Bacillus spp.?

Answer: Revised throughout the text except for the genus name.

_ L.493 (Conclusions): This entire section should be improved and focus on the results obtained. Nothing was mentioned about the promotion of growth in sugar beet plants by the bacterial isolates. Please improve the Conclusions accordingly.

Answer: We agree, and we have adjusted it accordingly.

-L.504: sp. not italicized. Also review the entire text.                    

Answer: Changed.    

Round 2

Reviewer 1 Report

Comments and Suggestions for Authors

The authors have significantly improved the manuscript, but several comments remain uncorrected (and apparently not understood)

1) What do the letters a, ab, de and others mean on the graphs Fig. 1A, Fig. 1B?

The authors added text in the caption to the figure “Different letters in bars (a, b, bc, etc.) denote 270 significant difference according to Duncan test (P < 0.05)”, but this did not make it any clearer, WHICH DIFFERENCES BETWEEN SAMPLES are we talking about?

Exactly, which differences do these letters represent? This needs to be made clear in the article text and graphics.

2) Line 22 - Should 16S rDNA be used? // the rDNA designation is incorrect; the correct spelling is "gene of 16S rRNA"

The article still contains multiple typos and a professional English speaker's editing of the manuscript is still recommended.

Comments on the Quality of English Language

The article still contains multiple typos and a professional English speaker's editing of the manuscript is still recommended.

Author Response

Dear Reviewer,

Thank you again for your helpful comments, which were considered in this revision.

Here are point-by-point responses to your queries:

1) What do the letters a, ab, de and others mean on the graphs Fig. 1A, Fig. 1B?

The authors added text in the caption to the figure “Different letters in bars (a, b, bc, etc.) denote 270 significant difference according to Duncan test (P < 0.05)”, but this did not make it any clearer, WHICH DIFFERENCES BETWEEN SAMPLES are we talking about?

Exactly, which differences do these letters represent? This needs to be made clear in the article text and graphics.

Response: We have removed "270" and revised the legend as follows:: 

'Different letters in bars (a, b, bc, etc.) denote a significant difference between the percentages of inhibition of the tested isolates according to the Duncan test (P < 0.05)'.

2) Line 22 - Should 16S rDNA be used? // the rDNA designation is incorrect; the correct spelling is "gene of 16S rRNA"

Response: This has been corrected throughout the text.

The article still contains multiple typos and a professional English speaker's editing of the manuscript is still recommended.

Response: We have utilized the editing service provided by the journal to polish the English language.

Thank you for your inputs.